

# Characteristics of Convective Snow Bands in the Baltic Sea Area

Julia Jeworrek[1], Lichuan Wu[1], Christian Dieterich[2], Anna Rutgersson[1]

[1]Department of Earth Sciences, Uppsala University, Uppsala, 75236, Sweden
[2]Swedish Meteorological and Hydrological Institute, Norrköping, 60176, Sweden

*Correspondence to*: Anna Rutgersson (anna.rutgersson@met.uu.se)

**Abstract.** Convective snow bands develop in response to a cold air outbreak from the continent over the open water surface of lakes or seas. The comparatively warm water body triggers shallow convection due to increased heat and moisture fluxes. Strong winds can align with this convection into wind-parallel cloud bands, which appear stationary as the wind direction remains consistent for the time period of the snow band event, delivering enduring snow precipitation at the approaching coast.

The statistical analysis of a dataset from an 11-year high resolution atmospheric regional climate model (RCA4) indicated 4 to 7 days a year of moderate to highly favorable conditions for the development of convective snow bands in the Baltic Sea region. The heaviest and most frequent lake effect snow was affecting the regions of Gävle and Västervik (along the Swedish east coast) as well as Gdansk (along the Polish coast). However, the hourly precipitation rate is often higher in Gävle than in the Västervik region. Two case studies comparing five different RCA4 model setups have shown that the Rossby Centre

atmospheric regional climate model RCA4 provides a superior representation of the sea surface with more accurate SST values when coupled to the ice-ocean model NEMO-Nordic as opposed to the forcing by the ERA-40 reanalysis data. The refinement of the resolution of the atmospheric model component lead especially in horizontal direction to significant improvement on the representation of the mesoscale circulation process as well as the local precipitation rate and area by the model.

*Keywords:* Baltic Sea, regional climate system modelling, extreme precipitation, shallow convection, SST

## 1 Introduction

The roughness and temperature differences between land and water surfaces often lead to local sub-climates such as mesoscale circulation systems or stable and unstable periods. For instance, during autumn and winter when the water surface temperature is still warmer than the average air temperature an unstable stratification develops in low levels since the ice-free water surface

appears as a source of moisture and heat to the overlying air mass. As a result, mesoscale convective precipitation events like convective snow bands may develop.

Convective snow bands are also known and studied as lake effect snow (at the Great Lakes), cloud streets, horizontal convective rolls or vortices with solid precipitation. At the Swedish east coast, they are often referred to as a snow canon ('snökanon'). Convective snow bands develop commonly over the open water surface of lakes or seas when cold air approaches

from the continent. Enhanced heat and moisture fluxes from the comparatively warm water body trigger shallow convection



as the colder air mass travels across the sea. An unstable atmospheric boundary layer builds up and the formation of shallow convective clouds is favoured. Relatively strong wind can organize this convection into wind-parallel quasi-stationary cloud bands with moving individual cells. Depending on various factors such as the strength of the horizontal wind, the vertical wind shear or the shape of the coast (Andersson and Gustafsson, 1993), different snow band structures can form (Niziol et al., 1995).

When the prevailing atmospheric conditions imply a strong development of convective snow bands, intense precipitation occurs locally where the snow bands hit the coast. The topographic changes from sea to land involve additional convergence and orographic lifting, which intensify the snowfall further. Since the circulation is organised in steady bands along the wind direction, usually only a limited area is affected by the snowfall. Thus, a reliable wind field forecast is crucial for a prediction of the hazard area.

Convective snow bands lead repeatedly to severe precipitation events in the cold season of the high mid-latitudes around the world. This phenomenon occurs frequently at several regions situated in high latitudes, including the Great Lakes (Kelly, 1986). However, the Swedish east coast experiences convective snow band events from the Baltic Sea several times a year. The large amounts of snow along with strong wind speeds can cause serious problems for traffic, infrastructure and other important establishments of society. Due to global warming it is in general expected that larger areas of lakes and inland seas

will stay longer ice-free (IPCC, 2015). As a result, the occurrence of those convective snow band events may become more frequent as well as more intense.

In this study an investigation was carried out on the ability of the model RCA4 (Rossby Centre regional atmospheric model) to simulate those snow bands, the sensitivity of the results to model resolution and surface forcing and to derive climate statistics of the occurrence.

## 2 Characteristics of Convective Snow Bands

Studies by Evans and Wagenmaker (2000), Niziol et al. (1995), Andersson and Gustafsson (1993) and Niziol (1987) showed that convective snow bands occur under specific conditions. The most important element for the formation of snow bands is the thermal difference between the water surface and the overlaying air (Mazon et al., 2014), which determines the extent of the essential heat and moisture fluxes. Therefore, it is necessary that a large part of the water surface is ice-free in order to

ensure sufficient sensible heat release and evaporation from the sea. Another important condition is the presence of instability to trigger convection. For the Great Lakes it has been observed that the minimum temperature difference between the water surface and the 850hPa level must be 13°C to initiate convective snow bands without additional synoptic scale forcing (Holroyd, 1971). This vertical temperature gradient matches approximately the dry adiabatic lapse rate. A larger value for the atmospheric lapse rate implies the presence of an absolute unstable layer within the lowest 850hPa. A big thermal difference

may furthermore enhance the moisture flux towards the air mass and support the formation of clouds and precipitation. The conditions for an intense development of convective snow bands are more favourable with increasing instability. However, the convection is vertically restricted. A capping subsidence inversion usually determines the height of the unstable boundary layer. This convective layer should extend at least 1 km above the surface in order to allow adequate convective cloud growth.





Nevertheless, Niziol et al. (1995) indicate that large heat and moisture fluxes from the water surface can significantly lift and even erode the inversion layer.

The wind field throughout the boundary layer plays an essential role for the evolution of convective snow bands. The most common and severe snow bands are aligned parallel to a strong prevailing wind (type I snow bands defined by Niziol et al., 1995) larger than 10 m s$^{-1}$ (Andersson and Nilsson, 1990). It should be noted that lower wind speeds of less than 5 m s$^{-1}$ may also lead to the development of shoreline-parallel cloud bands, initiated by a thermally driven land-breeze circulation (type IV snow bands defined by Niziol et al., 1995). A combination of wind speed and wind direction, and thus the distance and path that the air travels across the water surface, determines how much time an air mass of certain properties has to absorb heat and moisture from the water. A longer fetch allows a stronger development of the snow band. Laird et al. (2003) found for idealised cases that cloud bands form when the ratio between the wind speed and the fetch distance over the open water is between 0.02 and 0.09 m s$^{-1}$ km$^{-1}$. Accordingly, for a wind speed of 10 m s$^{-1}$ the fetch distance has to be between 110 and 500 km. Therefore, stronger winds require larger fetch distances.

The directional wind shear within the convective boundary layer is observed to be small for the time period of the snow band event. Niziol (1987) established a criterion for the likelihood of lake effect snow depending on the wind shear within the steering layer of the snow band (which determines approximately the first 50hPa above the ground up to 700hPa). Thus, convective snow bands are likely to occur for a directional wind shear of less than 30°. At a directional wind shear between 30° and 60° snow bands are possible to develop, however, beyond 60° the band like structure will break down.

Even the shape and the topography of the coast surrounding the water body can be essential for the snow band evolution. Andersson and Gustafsson (1993) investigated that the genesis areas for convective cloud bands are often bays at the 'coast of departure'. A convergence zone develops as two opposed land breezes meet in the centre of the sea. The secondary circulation system forces convection, which continues downwind as bands of convective rolls. When reaching the 'coast of arrival', the convection can be enhanced by another land breeze raising the capping inversion. Snow bands tend to organise themselves parallel to a concave shaped shoreline. Islands that are located along the fetch disturb the heat and moisture flux from the sea to the air mass locally and can cause multiple bands to reorganize and merge or split up.

## 3 Numerical Model Systems

### 3.1 RCA4

The Rossby Center of the Swedish Meteorological and Hydrological Institute (SMHI) is developing and applying climate models since 1997 (Jones et al., 2011). RCA4 is the latest version of their regional atmospheric climate model and it is running over many different Coordinated Regional Climate Downscaling Experiment (CORDEX) domains (Nikulin, 2013). The domain used in this study (illustrated in red in Fig. 1) covers Europe. RCA is based on the operational numerical weather prediction model HIRLAM, although, RCA was developed to simulate based on climatological time scales. The foundation of this hydrostatic model are primitive equations using terrain-following hybrid vertical coordinates and a rotated longitude-latitude-grid. The original model employed time steps of 15 minutes, 40 model layers as vertical coordinates and a spherical



resolution of 0.22°, which is corresponding to about 25 km horizontal grid spacing (Dieterich et al., 2013). Initial and lateral conditions for parameters like ice cover, sea surface temperature (SST) or wind speed are provided to the model every six hours by the interpolated ECMWF reanalysis data ERA-40 (Uppala et al., 2005).

### 3.2 RCA4-NEMO

The atmosphere-ocean interaction is of great importance for the atmosphere's properties and dynamics as well as the entire earth's climate system. The SST is a crucial factor for the accurate representation of the development of convective snow bands. The Nucleus for European Modelling of the Ocean NEMO (Madec, 2012) is an ice-ocean model based on primitive equations. Its domain in the present study covers the North Sea and the Baltic Sea and as seen in blue in Fig. 1, the model setup used here is denoted NEMO-Nordic. The boundaries at the northern North Sea and the English Channel are kept open

and take information of the Atlantic Ocean outside the NEMO-Nordic domain into account (Dieterich et al., 2013). In comparison with the default settings of the RCA4 model, NEMO-Nordic has a very high resolution with a horizontal grid spacing of two nautical miles, corresponding to circa 3.7 km, and 56 geopotential levels at the vertical scale (Dieterich et al., 2013). NEMO-Nordic can be coupled to RCA in order to exchange information at the interfaces between air and sea or ice. The ice-ocean model provides parameters such as ice fraction and albedo as well as SST to the atmospheric model. In turn

RCA4 communicates heat, freshwater and momentum fluxes to the NEMO-Nordic model (Wang et al., 2015). The coupling of two independently developed model components such as RCA4 and NEMO-Nordic can be realized by OASIS3 - the Ocean Atmosphere Sea Ice Soil Simulation Software. This coupler was developed by PRISM, the Project for Integrated Earth System Modelling and is commonly used in the climate modelling community (Valcke, 2013).

### 3.3 RCA4-WAM

Waves have an impact on the roughness length at the water surface, affecting in return the low level wind field as well as the heat fluxes. RCA4 has been used in connection to a wave model in order to test the sensitivity of this interrelationship. The WAve Model (WAM) is a third generation full-spectrum prognostic wave model using the basic transport equation (WAMDI Group, 1988), which can be used for an atmosphere-wave coupled system. The WAM model explicitly solves the energy balance equation in order to gain the evolution of the wave spectrum (Janssen, 2004). The European Centre for Medium-Range

Weather Forecasts (ECMWF) is running a coupled system of WAM in communication with an atmospheric component operationally since 1998 (ECMWF, n.d.). For the purpose of coupling, WAM and RCA4 have the same resolution and time step frequency in this study. Here the WAM model component is treated as a subroutine which is called by RCA4 with every time step communicating the essential information between the model components. The WAM model provides RCA4 with wave information in exchange for wind field data from the atmospheric model. The important wave data for the RCA model

is obtained by a two-dimensional ocean wave spectrum and may involve parameters like wave height and period as well as roughness length. The applied coupled RCA4-WAM setup is similar to that described by Wu et al. (2015) and Rutgersson et al. (2012) only exchanging the roughness length.



## 4 Method and outline

In order to give a good representation of the Baltic Sea, regional models with high resolution are essential to reproduce topographical features and substantial processes. In connection with the investigation of the mesoscale processes determining convective snow band events, this study was carried out in two parts: The statistical analysis of snow band events based on an

11-year RCA4 dataset and the evaluation of the use of different regional climate model systems.

First, the atmospheric regional climate model RCA4 was used in a high resolution to simulate the atmosphere over the 11-year time period from 2000 to 2010 with a spin-up of two months. The horizontal RCA4 resolution was set to 0.16°, which corresponds to approximately 18 km grid spacing. In order to keep the model numerically stable, the time step is 10 minutes. Based on the criteria listed in Sect. 2, days of convective snow band conditions were selected and statistically analysed with

respect to the season and the strength of the snowfall. The applied criteria are summarized in Table 1 and distinguish with different threshold values between moderate and favourable conditions for snow band development. The investigation is focused on convective snow bands which develop over the Baltic Sea and lead to snowfall at the Swedish coast. The criteria were therefore applied to either the Baltic Sea area as seen in Fig. 2a or the specified precipitation sector as seen in Fig. 2b.

In the second part of this study, two case studies are presented, comparing the atmospheric properties simulated by five

different model setups in order to give an assessment of the specific model performance concerning snow bands. An overview over the five model systems is given in Table 2. The regional atmospheric climate model RCA4 has been used by itself in the original resolution of 40 vertical model layers and 0.22° (about 25 km) horizontal grid spacing with a spin-up time of approximately two months ahead of the snow band event. A coupled simulation of the RCA4 with the ocean model NEMO-Nordic was carried out to investigate the impact of the SST. In order to provide enough time for adjustment between the

models, a spin-up of almost two years was used. The two components in the coupled system of RCA4 and WAM are identical with regards to horizontal resolutions and time steps, while WAM provides the RCA model with a sea surface roughness corresponding to the atmospheric wind field above. The final two experiments were performed using RCA4 simulations with increasing resolutions either in horizontal spacing or in vertical direction. The horizontal resolution was refined from 0.22° to 0.11° (about 12.5 km) and the 40 model layers of the original RCA4 set-up were increased in a different run to 62 layers. The

spin-up time was used similar to the RCA simulation. The higher resolution, however, requires a lower time step, which was therefore decreased from 15 to 5 minutes.

## 5 Results

### 5.1 Analysis of an 11-year RCA4 dataset

The atmospheric conditions which favour the development and maintenance of convective snow bands show a recurrent pattern

in the Baltic Sea region. In order to select a day as convective snow band event all of the following criteria must be fulfilled within the respective area (defined as in Fig. 2): The maximum 10m wind speed over the Baltic Sea area must be larger than 10 m s$^{-1}$; the mean 2m temperature over the Baltic Sea is required to be smaller than 5°C for favourable conditions and smaller



than 8°C for moderate conditions; the maximum temperature difference between the sea surface and the 850hPa level over the Baltic Sea area, which determines the instability, has to be greater than 13°C for moderate conditions and greater than 15°C for favourable conditions; the mean wind shear over at least 50% of the Baltic Sea area between the levels of 700hPa and 975hPa must be smaller than 60° for a possible snow band development and smaller than 30° for a likely presence of snow

bands; the mean wind direction at 900hPa over the Baltic Sea area has to be in the range of 0° to 90° (north to west wind); the maximum boundary layer height has to be larger than 1 km somewhere over the Baltic Sea area; the maximum precipitation within the precipitation area must be larger than a rate of 0.5 mm h$^{-1}$ for moderate conditions and larger than 1 mm h$^{-1}$ for favourable conditions; and finally the maximum snowfall rate of the precipitation area has to be larger than 1.5 mm d$^{-1}$ as well as larger than 0.5 mm h$^{-1}$ for favourable conditions (see also Table 1).

The high resolution RCA4 simulation from 2000 until 2010 reproduces a total of 121 days with these criteria fulfilled. 49 of these days are within the limits of the stricter thresholds describing favourable conditions for snow band formation. Correspondingly, favourable atmospheric conditions occur for a strong development of convective snow bands over the Baltic Sea on average 4.5 days a year. When including the remaining cases which meet the weaker criteria, a total average of 11 days per year is obtained. Figure 3 displays the distribution of the cases per year and month. It is seen that the total number of days

varies between 5 and 22 per year. Most days occur in the months of November and December (Fig. 3b).

The maximum 10m wind speed of all selected snow band days is on average 13.3 m s$^{-1}$. The mean wind direction varies between all cases approximately from 0° to 65°. Hence, north and north-west wind is most common. Westerly wind, which could generate snow bands from the Gulf of Finland (Mazon et al., 2015), is surprisingly not seen for any period. The mean wind shear over half of the Baltic Sea area is small for all days. Most cases represent mean wind shear values of even less than

10°. The maximum temperature gradient of the lowest 850hPa is on average 18°C and the maximum boundary layer height over the Baltic Sea is usually around 1.7 km.

The largest impact of convective snow bands on the public life is due to the heavy snowfall. The intensity of a snow band event may be defined by the consequent amount of its precipitation at the coast. The scale of the given snowfall values refers to the volume that the snow would possess in liquid, rather than solid form. As an approximate relationship it is reasonable to

assume that one millimetre of melted snow corresponds roughly to one centimetre of gained snow depth, although the density of the snow depends strongly on parameters like temperature and age of the snow (Dubé, 2003). The selected days indicate a maximum snowfall rate between 0.2 and 3 mm h$^{-1}$. While most values were smaller than 2 mm h$^{-1}$, only two outliers in January and February 2007 attained values of approximately 2.6 and 2.9 mm h$^{-1}$. The mean value of the maximum snowfall rate of each day amounts to 0.8 mm h$^{-1}$. Accumulated over the day, the snowfall varies between 1.5 and 17 mm d$^{-1}$, with a total mean

of 5.8 mm d$^{-1}$.

The regions affected by the precipitation of convective snow bands in the Baltic Sea area can be seen in Fig. 4. This data represents the snowfall as an average value of all selected days as well as only for the days which meet the criteria for favourable conditions. The precipitation reference sector considered for the selection criteria (as defined in Fig. 2b) is framed in black. However, the figures represent a sector that exceeds the precipitation area in order to observe whether other regions





are also affected by enhanced precipitation on the same days. The average snowfall rate results from the total snowfall of the selected days (of favourable or favourable and moderate atmospheric conditions for convective snow bands) normalized by the number of days with positive snowfall at a specific location. Accordingly, along the Swedish coast two separate regions are mainly pronounced by a large average daily precipitation rate. The concavely shaped shore near the Swedish town Gävle

represents the most intense snowfall within the precipitation sector with up to 3 mm d$^{-1}$ on average for the days of favourable conditions. The area around the town Västervik at the Swedish east coast, west of the island Gotland, also displayed a large area of high average snowfall at around 2.5 mm d$^{-1}$. In addition, the Swedish island Gotland indicates pronounced precipitation. Outside of the precipitation sector another hotspot appears in the Gdansk region at Poland's north coast. The area of high average snowfall reaches values up to 2.5 mm d$^{-1}$. Around the Baltic Sea it can be observed that all coasts which are facing

north, northwest or west experience enhanced snowfall for the selected snow band days.

A similar picture is obtained regarding the frequency of favourable conditions for convective snow bands. While the selection criteria for favourable conditions were fulfilled 4.5 times a year on average, approximately once per year the Gävle region received snowfall rates larger than 5 mm d$^{-1}$ (Fig. 5). Snowfall of more than 10 mm d$^{-1}$ due to convective snow bands occurs approximately every third year. The hourly precipitation rate is often higher in Gävle than around Västervik (Fig. 6). Although

lower hourly snowfall rates between 0.5 and 1 mm h$^{-1}$ occur in the Västervik region more often (about every 8 months) and cover a larger region than in Gävle (where they occur about once a year), higher snowfall rates of larger than 1 mm h$^{-1}$ are more frequent in Gävle (approximately once every one and a half years) than in Västervik (where this happens only every third year). Outside of the precipitation sector the area near Gdansk indicates snowfall larger than 5 mm d$^{-1}$ about every one and a half years.

**5.2 Case studies**

Two different cases have been selected to study the sensitivity of the model setup. Five different RCA4 based model setups are used (Table 2) considering the atmospheric conditions of convective snow bands. Case 1 concerns a well-studied event in early December 1998 that caused 130 cm solid snowfall within three days in the Swedish town Gävle (SMHI, 2015). The less intense case 2 occurred in early February 2001 and had similar synoptic conditions. Cold air was transported from Finland

over the Gulf of Bothnia causing extreme snow precipitation at the Swedish coast close to Gävle. While the wind direction in case 1 remained consistent over several days by accumulating snowfall at a restricted area around Gävle, in case 2 the wind direction turned slightly and distributed the precipitation more along the Swedish coast. Both cases lasted three to four days and for a better understanding of the evolution of the atmospheric situation approximately one day ahead and one day after the snow band have been taken into account for the investigation.

All model systems show similar behaviour for the wind field development. The maximum 10m wind speed over the Baltic Sea increases rapidly with the beginning of the convective snow band event and slowly decrease in the following days. Due to the wave feedback through the wind-wave interaction as well as the nonlinear wave interaction in the 2 dimensional wave spectrum calculation of the roughness length in the WAM model this development showed in both cases a delay by several hours for the



RCA-WAM simulation relatively to the other models. The comparison with SMHI station measurements indicates that most models underestimated the 10m wind speeds at the Swedish coast. The best representation of the observational wind data has been obtained by the RCA model with increased horizontal resolution. The mean wind shear between 975hPa and 700hPa was in both cases around 30° for the considered area of the Gulf of Bothnia. Case 2 indicates higher wind shear values for the days

before and after the snow band event.

The air temperature field indicates for all models clearly that the Baltic Sea serves as a heat source to the air above. But due to the approaching cold air mass the temperature over the Gulf of Bothnia decreases rapidly. In case 1 the maximum 2m temperature decreases within two days by 4°C, in case 2 it decreases even by 8°C. Between all models RCA4-NEMO simulated systematically higher 2m temperatures, owing to a higher SST.

For the high resolution RCA setups as well as the RCA-WAM the ice cover and SST is provided to the RCA model by the ECMWF reanalysis data ERA-40. The spectral resolution of ERA-40 is T159, which corresponds to 1.125 degrees or approximately 125 km (Advancing Reanalysis, 2016). Hence, the sea surface input based on ERA-40 has a coarser resolution than the original RCA itself. Alternatively, the ice-ocean model component NEMO-Nordic, simulates its own SST in a much higher resolution (see Sect. 3.2) and can represent local features in more detail. When comparing the SSTs between ERA-40

and NEMO-Nordic interpolated to the original RCA resolution, the NEMO-Nordic SST resulted in generally higher values for both cases (see Fig. 7 for case 2). Only in the straits where the Baltic Sea is shallow, the SST difference between ERA-40 and NEMO-Nordic is small and at some locations even positive. However, the region of interest, the Gulf of Bothnia, shows in both cases significant differences. NEMO-Nordic obtained up to 5°C higher SSTs compared to ERA-40. The OISST data (NOAA, n.d.), which is result of the combination of measurements from satellites, ships and buoys indicated better agreement

with the NEMO-Nordic than the ERA-40 data. In the Gulf of Bothnia, the NEMO-Nordic SST was in some locations around 1.5°C warmer than the OISST where the ERA-40 SST is up to 4°C colder. As well, the development of sea ice cover is better represented by the NEMO-Nordic model.

The SST is furthermore associated with the heat fluxes over the water surface. Accordingly, it is not surprising that the RCA4-NEMO model represents the highest heat fluxes among the models for both, sensible and latent heat, over the Gulf of Bothnia

(Fig. 8). However, both case studies show that the RCA4-NEMO heat fluxes agree well with all other models before the snow bands arise. The difference develops with the initial occurrence of the snow bands. All models represent an increasing sensible and latent heat flux development throughout the convective snow band event. The maximum heat fluxes are reached at a later stage of the evolution when the cloud bands start to dissolve again. The magnitude of the heat fluxes in the February case has been larger than in the December case. Similar to the temperature change, the evolution for the heat fluxes was also more rapid

in case 2.

The increased wind speeds and heat fluxes during the convective snow band event also causes the boundary layer height to rise. Based on the investigation of the two cases it has been observed that RCA4-NEMO represents comparatively high boundary layer heights, while the RCA model with high horizontal resolution tends to give a shallower boundary layer (see



Fig. 9). Although case 1 was an intense event the mixed layer height barely exceeds 1 km, which has been defined as a threshold by the criteria in Sect. 2.

Depending on the strength of the convective snow band the amount of the total precipitation can vary vastly. A slight turning of the wind may furthermore lead to a distribution of the lake effect snow along the coast rather than the accumulation in one

restricted area. Case 1 indicated a significantly greater accumulation of precipitation than case 2, partly due to the consistent wind direction. The hourly precipitation of case 1 has however been more intense for a longer period of time. Regardless, for both cases the high resolution RCA models as well as the RCA4-NEMO system indicate considerably higher precipitation rates than the original RCA or RCA-WAM. The 48h accumulated total precipitation of case 2 reached up to 20 mm and can be compared for the different models in Fig. 9. In case 1 the precipitation of 2 days went even up to 45 mm. Very remarkable

were the results of the RCA simulation with increased horizontal resolution. The local maxima reach significantly higher values and the confined precipitation area is represented in more detail. When comparing the model performance for the location of a weather station in Gävle with the measured precipitation, as in Fig.10, it is clear that all models have difficulties in representing the exact time and location of the snowfall. For case 1, almost all models underestimate the actual precipitation. Only the RCA model with high horizontal resolution exceeds the measurements for one day. In case 2, on the other hand, all

models represent maximum precipitation rates at a time period that did not show any precipitation in the measurement data, while the measured precipitation showers in Gävle were not recognized by the models.

## 6 Discussion

The simulations by RCA and RCA-WAM resulted in a rather weak development of the convective snow bands and the comparison with observational data showed that the atmospheric conditions were often underestimated by these two models.

The coupling of the atmospheric RCA model with the wave model component WAM employed a different roughness length computation at the sea. Hence, the largest impact was observed on the wind field. The magnitude of the maximum 10m wind speed reaches similar values, however, the RCA-WAM model possesses a time shift due to the wave feedback on the roughness length and wind speed. With regard to all other investigated parameters RCA and RCA-WAM show very similar results and the different roughness length calculation due to the coupling of RCA and WAM has a negligible impact on the atmospheric

conditions describing convective snow band events.

The coupled atmosphere-ocean system RCA4-NEMO benefited clearly from the high resolution SST of the NEMO-Nordic component. The reanalysis ERA-40 data which otherwise provided the uncoupled RCA model with the SST had a coarser resolution than the original RCA itself. For water bodies of the size as the Baltic Sea, this resolution is insufficient and the quality of the simulation is impaired as local extremes cannot be represented. The interpolated ERA-40 SST shows a negative

bias towards the NEMO-Nordic SST as well as independent datasets such as OISST. The difference is significant with up to 5°C between ERA-40 and NEMO-Nordic. While the NEMO-Nordic simulated SST was shown in some locations to be slightly higher than the OISST data, it provided a better representation of the sea surface properties than ERA-40. The higher SST of the RCA4-NEMO model caused furthermore larger heat fluxes and it increased the instability through a larger temperature



difference within the lower layers resulted in an enhanced convection and a higher boundary layer height and finally, larger local precipitation rates.

A high resolution is of great importance when it comes to the regional modelling of mesoscale high impact events. Increasing the resolution of the atmospheric RCA model resulted in a great improvement for the model performance. Both high resolution
simulations indicate larger values for the local maximum of the 10m wind speeds over the Gulf of Bothnia. The best agreement with observational data, however, was obtained by the RCA model of increased horizontal resolution. Although, the temperature and heat fluxes did not show any impact from an increased resolution in any direction, the high horizontal resolution RCA was able to represent the mesoscale atmospheric circulation process associated with convective snow bands in more detail and resolved the local precipitation rate and area more precisely when comparing to the other models.

Finding a method to select convective snow band events is not straight forward. Even though the atmospheric conditions are typical and can be described by certain criteria, the thresholds applied imposed a larger impact on the number of days which are selected. Since convective snow bands arise from cloud bands that may initially not have any precipitation, it is not always clear how to define the transition and where to set the threshold for heavy precipitation due to snow bands. The average precipitation did not appear very intense, as some days have been selected with rather low snowfall rates of snow bands which
could not have developed as strong. According to the results of the case studies, the RCA model often underestimates the exact amount of precipitation. The increased resolution improves the results on the representation of the snow bands, however, for a long dataset of 11 years it is very computationally intensive to run the simulations in such a high horizontal resolution was done as in the case studies. Also the use of the ERA-40 data as sea surface input has shown its drawbacks, and caused the RCA model to simulate snow bands in a less intense evolution. The exact amount of the snowfall and its return period should
therefore be interpreted with caution. However, the result for the distribution and the relative amount of the snowfall can be understood in a qualitative rather than a quantitative sense. The Gävle region possessed the largest average snowfall rates and the shortest return periods for comparatively high hourly snowfall rates as a result of snow bands developing over the Gulf of Bothnia. It seems natural that convective snow bands which require a cold air outbreak over the warmer sea will develop more frequently in the most northern part of the Baltic Sea. The shape of the Gulf appears furthermore conducive to the generation
of convergence zones for the initial formation and the bay-shaped coast in the Gävle region enhances the precipitation due to orographic forcing once again. The snow precipitation reaches occasionally well 100km inland, however, the most intense lake effect snow falls approximately within a radius of 50 km.

North-west winds lead convective snow bands also often to the Västervik region, however, the average snowfall here is not as intense nor as frequent. Convective snow bands occur often in multiple band structures and it is common that various regions
along the coast are affected by the lake effect snow when the atmospheric conditions are favourable in a large area. Hence, the Gdansk region was also perceived to experience convective snow bands on the same days as the Swedish coast.

Since convective snow bands occur in different strengths around the year, a statistical analysis as performed in the first part of this study must define the range of snow band criteria depending on their intensity. The present study includes snow bands



that caused moderate snowfall. However, in order to investigate stronger snow band events with hazardous consequences a longer period of time should be studied with stricter criteria for the precipitation since their occurrence it not as frequent.

## 7 Conclusions

5 The investigation of an 11-year RCA4 climate model dataset indicated the heaviest and most frequent lake effect snow due to convective snow bands in the Baltic Sea area affecting the Gävle region. The Västervik region at the Swedish coast and even the Gdansk area at the Polish coast also experienced enhanced snowfall on days of favourable atmospheric conditions for snow bands. When including days of moderate conditions for convective snow band development, a total of 11 days per year on average results. Most convective snow band events occur in the months of November and December, when the sea surface is
10 still warm from the summer and the cold air approaches frequently from the cold Finish land.

The application of RCA4 in different model setups has indicated for the two case studies that any of the investigated RCA4 model configurations simulate the atmospheric conditions for convective snow bands and fulfils the criteria established in previous research. Nevertheless, significant differences have been observed between the different model systems.

The RCA and RCA4-NEMO model varied largely due to the different SST input provided by coarse reanalysis data or the
15 high resolution ocean model respectively. The coupled RCA4-NEMO model provided a superior representation of the sea surface with significantly higher SST values when comparing with the ERA-40 data. The direct impact of the higher NEMO-Nordic SST on the heat fluxes and convective development manifests itself through a more intense convective snow band development and higher local precipitation rates. Even if on a larger scale all models agree well on the overall precipitation area, the exact location on a smaller scale as well as the amount and time of the snowfall remain a challenge. The models
20 differed to a great extent in the amount of accumulated precipitation. The largest precipitation rates were given by the two high resolution models as well as the atmosphere-ocean model.

Since the atmosphere ocean interaction is of great importance for the regional climate modelling of events like convective snow bands, the coupling with the high resolution ocean model NEMO-Nordic is advantageous over the use of the coarse reanalysis data used in the original RCA model. Moreover, the increased resolution of the atmospheric RCA model had a
25 positive impact on the model results. The high horizontal resolution lead to an especially significant improvement on the representation of the cloud bands, the precipitation area as well as the wind speed. Based on the investigation of the two cases the use of a coupled atmosphere-ocean system in connection with a high horizontal resolution of the atmospheric component is suggested for a more accurate representation of convective snow bands in regional climate models.

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



**Table 1**: Criteria for the selection of days with moderate to favourable atmospheric conditions for convective snow bands. The Baltic Sea area and precipitation area are defined in Fig. 2.

| Region | Parameter | Weak criteria | Strong criteria |
|---|---|---|---|
| Baltic Sea area | Max 10 m wind speed | > 10 m/s | |
| | Mean 2 m temperature | < 8°C | < 5°C |
| | Max temperature difference (between surface and 850 hPa) | > 13°C | > 15°C |
| | Mean wind shear (between 700 hPa and 975 hPa) of 50% of the Baltic Sea area | < 60° | < 30° |
| | Mean wind direction (at 900 hPa) | between 0° and 90° | |
| | Max boundary layer height | > 1000 m | |
| Precipitation area along the Swedish coast | Max precipitation | > 0.5 mm/h | > 1 mm/h |
| | Max snowfall | > 1.5 mm/d | |
| | | | > 0.5 mm/h |



**Table 2:** Overview of the model experiments used for the analysis of an 11-year dataset and the model evaluation based on two case studies.

|  | Experiments | Abbreviation | Horizontal resolution | Vertical levels |
|---|---|---|---|---|
| 11-year dataset | RCA4 with increased horizontal resolution |  | 0.16° | 40 |
| Case Studies | RCA4-NEMO | RCA4-NEMO | 0.22° | 40 |
|  | RCA4 | RCA | 0.22° | 40 |
|  | RCA4-WAM | RCA-WAM | 0.22° | 40 |
|  | RCA4 with increased horizontal resolution | RCA high horiz | 0.11° | 40 |
|  | RCA4 with increased vertical resolution | RCA high vert | 0.22° | 62 |





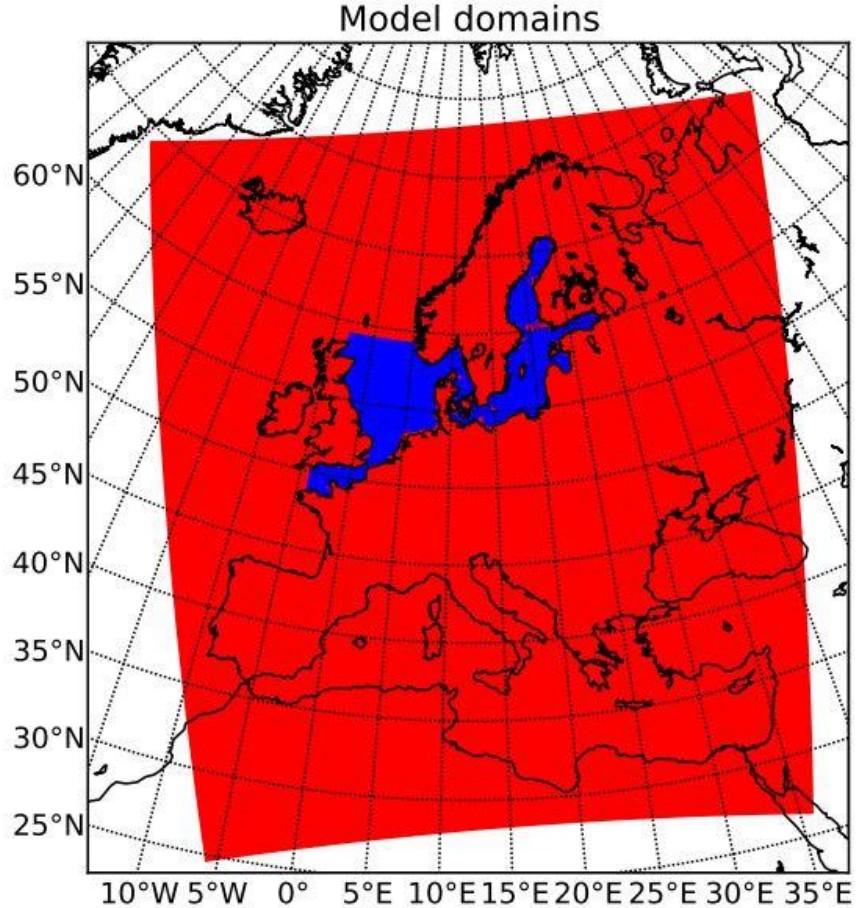

**Figure 1:** The domain of the NEMO-Nordic model (in blue) embedded in the RCA European CORDEX domain (in red). (Dieterich et al., 2013)





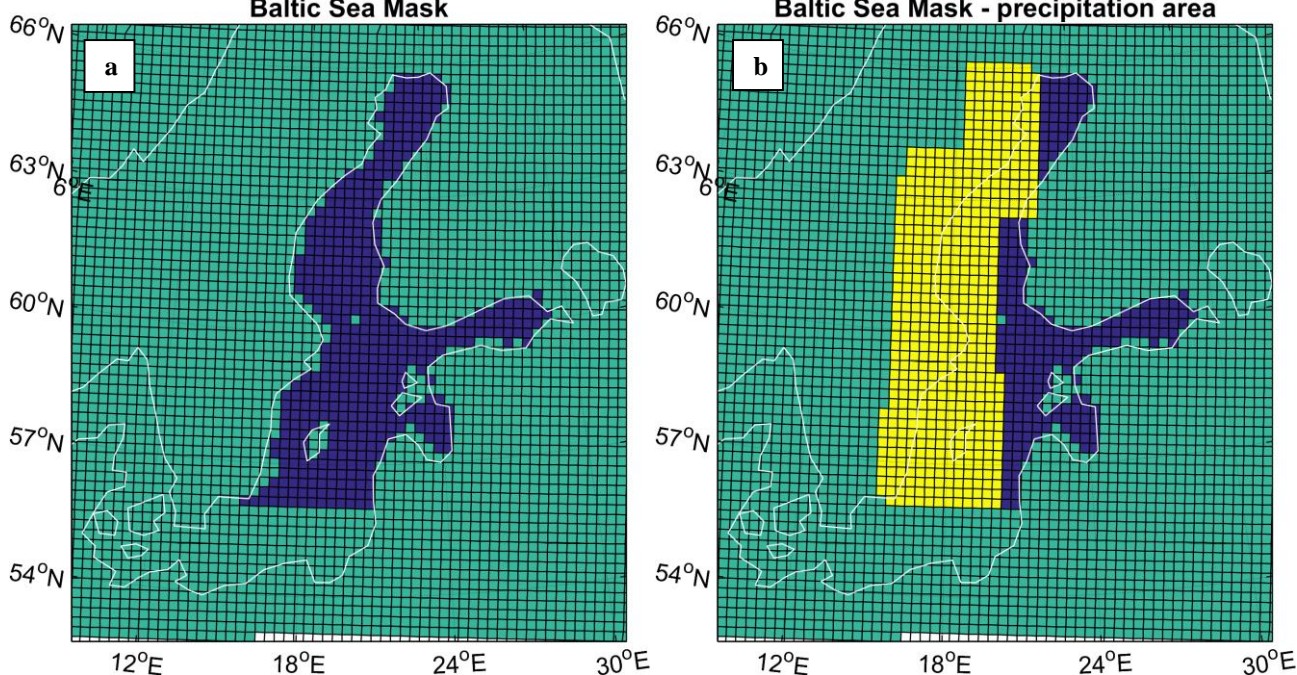

**Figure 2:** Baltic Sea area (in blue, **(a)**) and the precipitation area (in yellow, **(b)**) considered for the criteria for the selection of days with convective snow band conditions (compare to table 1).





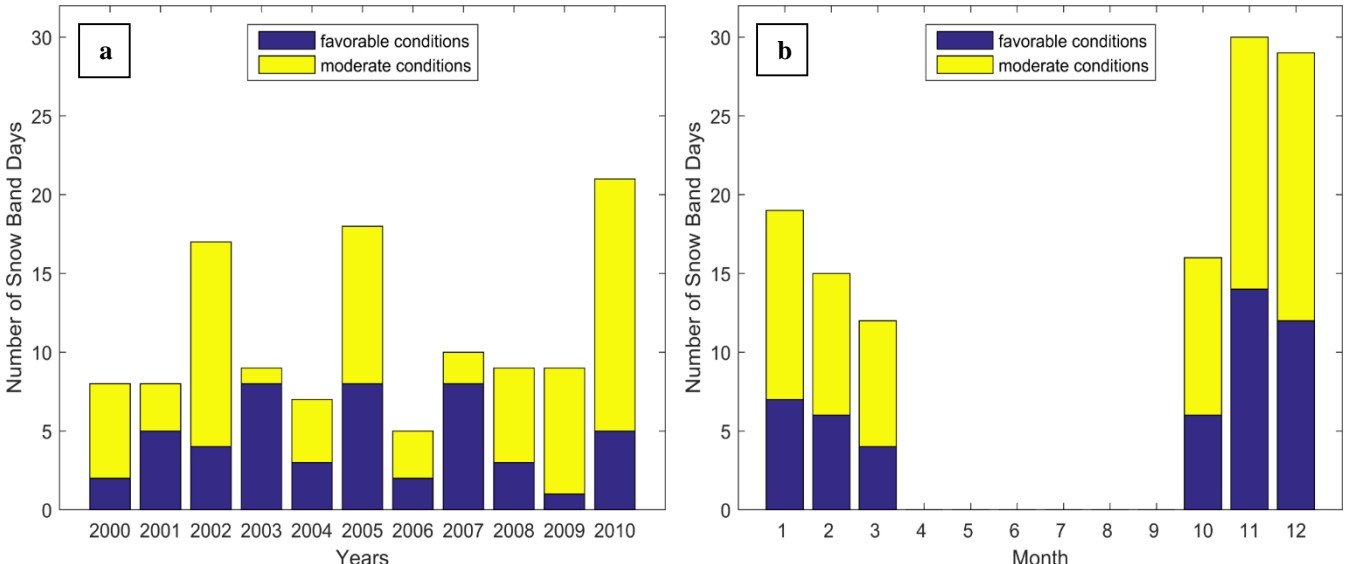

**Figure 3:** Number of snow band days fulfilling the applied criteria per year (a) and month (b) respectively.




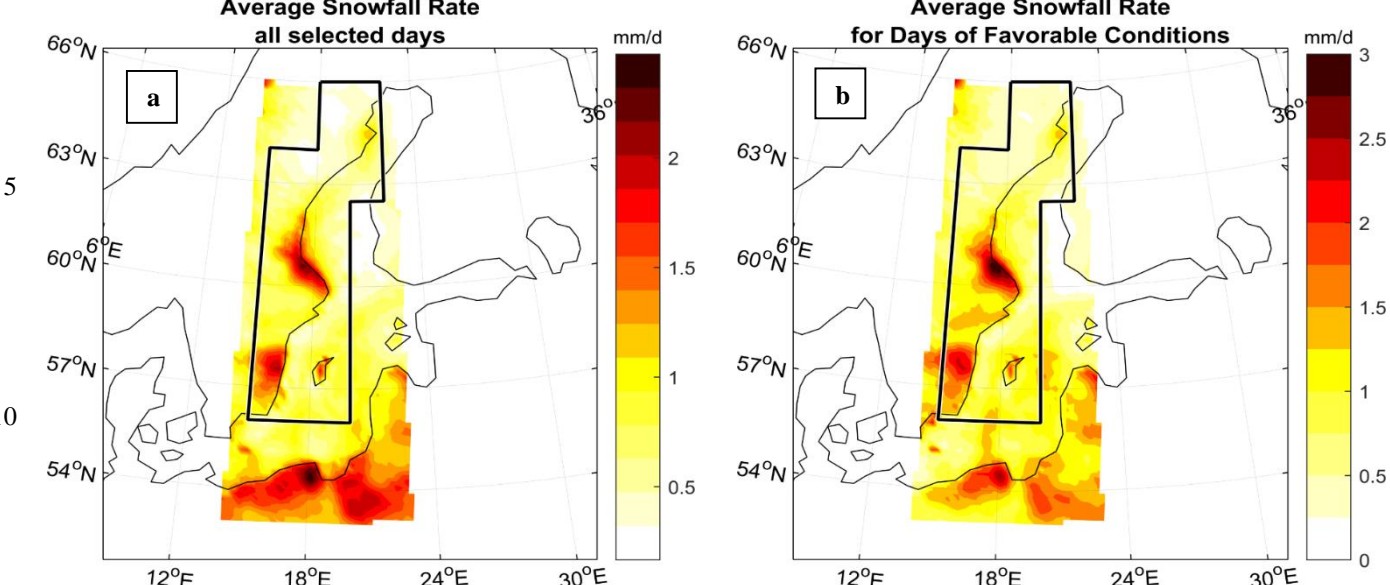

**Figure 4:** Average snowfall rate for all 121 selected days **(a)** and for the 49 days of favourable atmospheric conditions for convective snow bands **(b)** from 2000 to 2010 normalized by the frequency of positive snowfall.





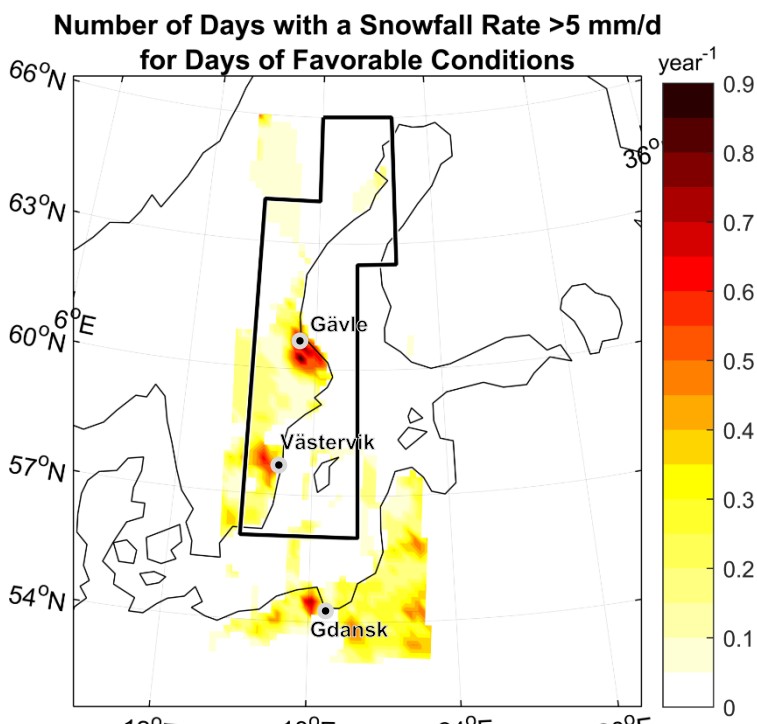

**Figure 5:** Frequency of occurrence for favourable atmospheric conditions with convective snow bands causing a snowfall rate later than 5 mm d$^{-1}$.



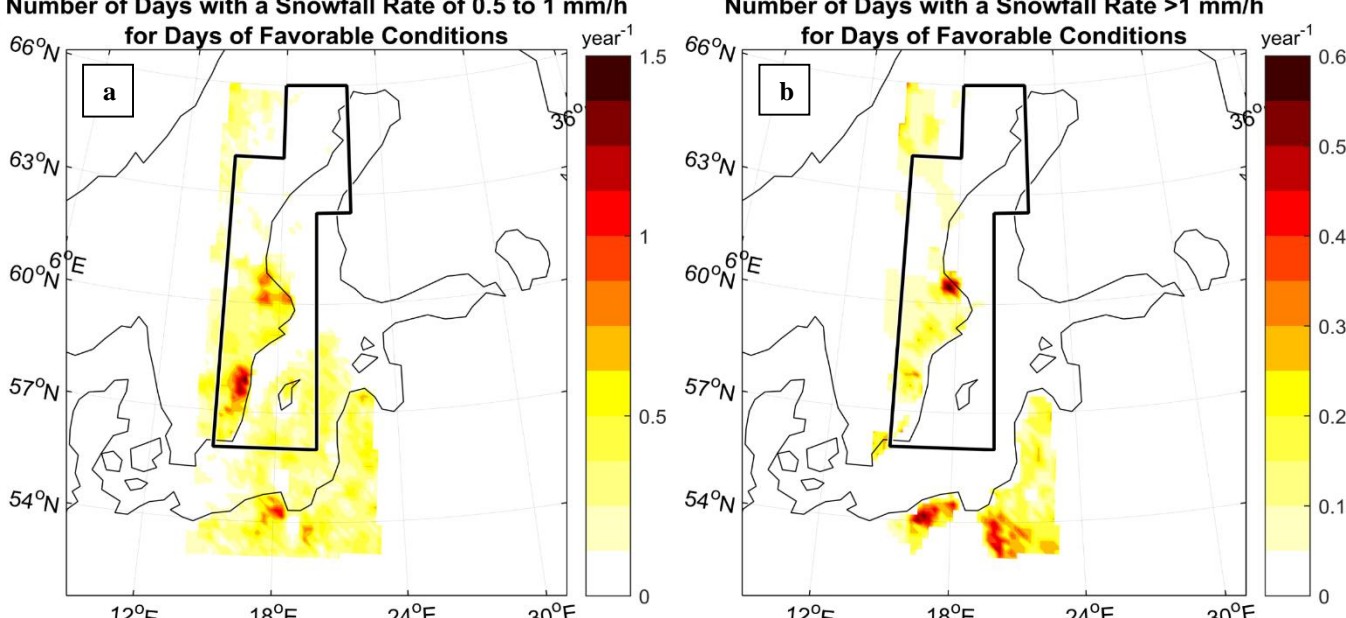

**Figure 6:** Frequency of occurrence of convective snow bands fulfilling the selection criteria for favourable atmospheric conditions (see Table 1) causing a maximum snowfall rate between 0.5 and 1 mm h$^{-1}$ **(a)** and larger than 1 mm h$^{-1}$ **(b)**.



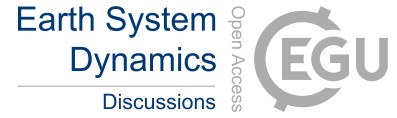

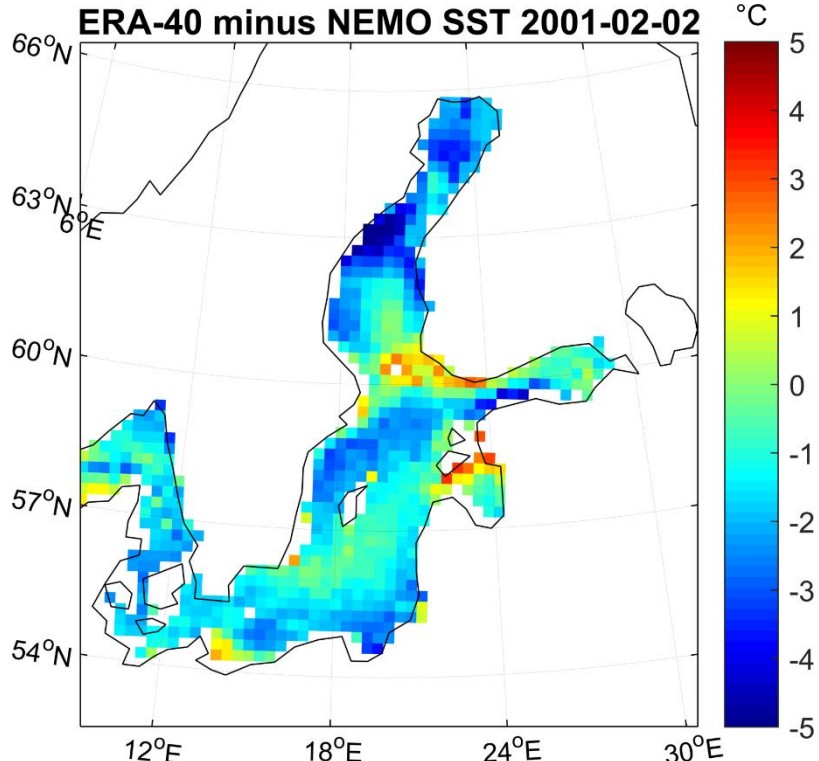

**Figure 7:** Difference map of the SST by ERA-40 and NEMO‑Nordic for one day of case 2.





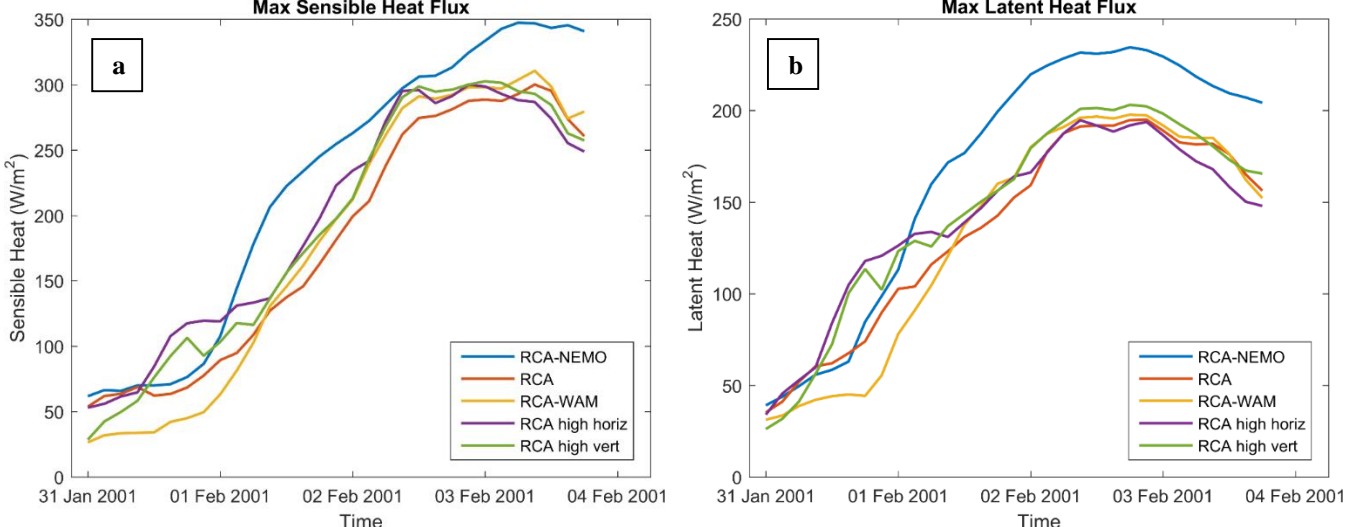

**Figure 8:** Maximum sensible heat flux (**a**) and maximum latent heat flux (**b**) over the Gulf of Bothnia of the time period of case 2.





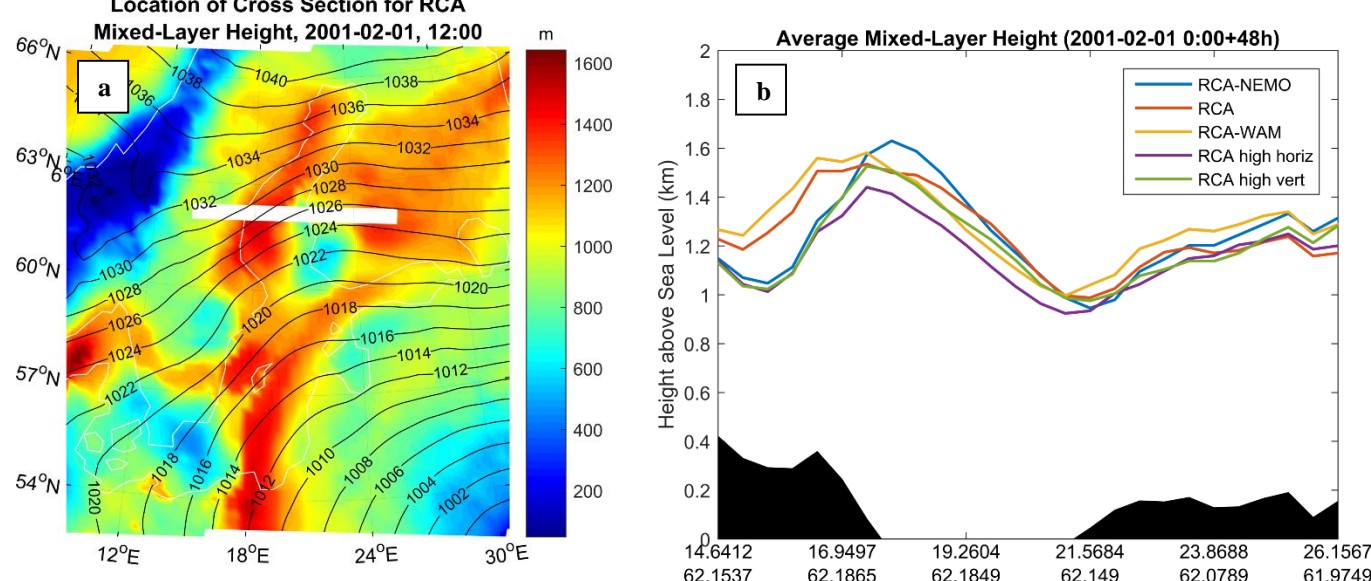

**Figure 9:** Boundary layer height horizontally for RCA (**a**) and the two-day mean in a cross section for all models (**b**) for case 2.



**Figure 10:** Two day accumulated total precipitation of all models for case 1 in comparison and a cropped satellite image of 1 February 2001 (MODIS, 2016).





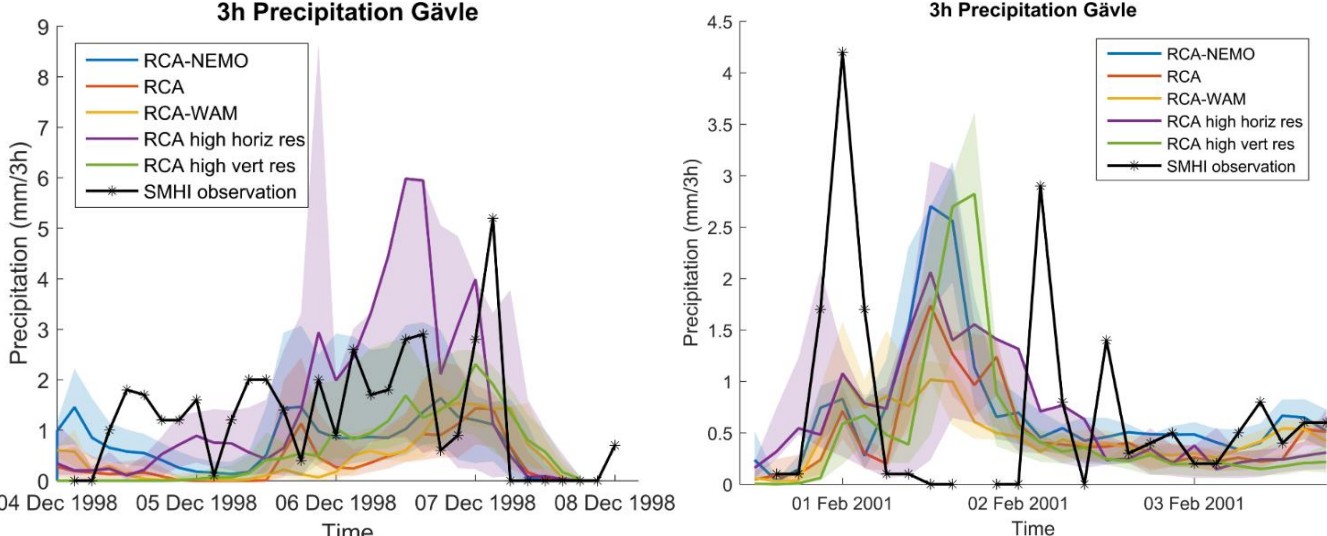

**Figure 11:** Time series of the precipitation measurements of the SMHI station in Gävle in comparison with the model results for this location. The solid line shows the model simulation at the closest grid point to the measurement site and the shaded area indicates the variation of this result according to the directly neighbouring grid points.