# Peer review of "Characteristics of Convective Snow Bands along the Swedish East Coast"

_Earth System Dynamics, 2016_

## Referee Comment (RC1) · Anonymous Referee #1 · 22 Nov 2016

The paper is devoted to 11-year climatology of convective snow bands in the Swedish east coast. Since convective snow bands bring high impact weather to coastal regions of the Baltic Sea, therefore it is important to study their characteristics and climatology. Former studies about the snow bands over the Baltic Sea area have been mostly devoted to understanding of the processes that cause the lake effect over the Gulf of Bothnia, or Gulf of Finland. Both are intensive areas of convection in late autumn or winter, during the cold air outbreaks at the open sea. This article is the first climatological study in this field for the Baltic Sea area. The paper is in many places unclear and I think that several points should be better addressed before considering it for a final publication.

1) The title is not exact, only snow bands that bring high impact weather to Swedish east coast are studied. Even in the introduction nothing is written about other Baltic Sea

coasts that also experience convective snow bands some times per year in average. It should be explained why these areas are let out of the scope of the paper. In the methods is written that the same criteria of detection were applied over the whole area of the sea. What is then the reason to leave other areas than Swedish east coast out? Is the resolution of the climate model not high enough to identify the snow bands also on the Finnish side of Gulf of Bothnia, at northern and southern coast of the Gulf of Finland or at the western coasts of Estonian islands. This kind of discussion should be included at least based on references.

2) The aim of the study is not clearly defined, I think it should be stressed that RCA4 is a climate model with not very high resolution. In earlier papers over this area the snow bands' formation processes have been studied with weather prediction models with much higher resolution. What is the reason to use climate model instead of weather prediction model? The conclusion that model with higher resolution can resolve the mesoscale circulations better is obvious. What is the novelty in this? What is the main new contribution?

3) It is misleading to write, that snowing takes place over the coast, it snows also over the sea (that could be followed also from the results). Andersson and Nilsson (1990) show that the highest precipitation is not at the coast, but close to it over the sea. This should be rewritten.

4) Showing the position of snow bands with just a satellite image (Fig 10) is not convincing, the high impact weather is connected to persistance of weather systems, the satellite image is valid just for the moment of the overflight. A radar image of 24 h or longer accumulation of the precipitation would be a good proof and source for intercomparison as from there approximate precipitation rates over the sea could be estimated and the regional distribution of the snow bands.

5) Very local characteristics have been chosen to identify the snow bands. Extremely high snow amounts are caused by weather systems that persist for a long time and

are supported by large scale atmospheric circulation. Did you check that in addition to the local favouring conditions to convection the large-scale conditions were fulfilled? I suggest to look these conditions in more detail at least in case of both case study periods.

6) The thresholds have been chosen based on references, mostly describing processes in other regions, more attention should be given to the studies that are devoted to the Baltic Sea region. References to several studies of Finnish scientists in this field are missing: Savijärvi et al 2012, where the same 1998 December Gävle storm is analysed, and Vihma, and Brümmer 2002, if to name some of them.

7) Do I understand right that Gdansk region in Poland comes out as a region where are strong snowfalls. How did you check it, that it is snow, not just rain? Was there the same temperature difference thresholds fulfilled as in the east coast of Sweden, or it just had high precipitation at the same time? Please give references that this kind of events happen often at this coast?

Specific comments:

Page 1 line 6 Cold air could come not only from the continent, but also from the ice covered sea.

Page 3 line 25, some general introduction to model systems is missing.

In Table 2 the logical order would be to begin the case study models with RCA and then go further with the more complex systems.

Was the same criteria applied to the whole Baltic Sea area or only to the region near the Swedish coast (p 5 line 13).

It is enough to present the criteria for detecting of snow bands just in Table 1, to write the same in the text is justified if some closer explanations are added. (p 5 line 31 and further).

How did these days distribute over the area geographically? (p 6 line 10 and further)

The precipitation amounts connected to these snow bands are not large or exceptional, even 17 mm per 24h is too low to call extreme. (the paragraph beginning at p 6 line 22)

Why only the precipitation for the sector shown in Fig 4 is presented? Are other data missing or the days selected could not be associated with precipitation in other regions?

The quality of the figures should be improved to add readability to this work

Fig 9 Coastlines could not be followed

Fig 10 Coastlines and isobars could not be followed

Fig 11 The texts in subplots are not with the same font, ticks are missing at x-axis.

---

## Referee Comment (RC2) · Anonymous Referee #2 · 28 Nov 2016

**Referee comments to the manuscript**

**Characteristics of Convective Snow Bands in the Baltic Sea Area**

**presented by Julia Jeworrek, Lichuan Wu, Christian Dieterich and Anna Rutgersson**

It is a nice study on climatology of convective snow bands in the coastal regions of the Baltic Sea focussing on the Swedish coast. The results are derived from the regional climate model outputs. The topic is interesting and important. This phenomenon is an extreme event that has brought serious problems for traffic and other human activities in nearly all coastal regions of the Baltic Sea. Unfortunately, convective snow bands are not well studied. In my mind, this manuscript could be evaluated well but it needs minor revisions. First, I'll answer to the general questions of the journal and then I'll make my more detail comments and suggestions.

1. Does the paper address relevant scientific questions within the scope of ESD? Yes.

2. Does the paper present novel concepts, ideas, tools, or data? Yes.

3. Are substantial conclusions reached? Yes.

4. Are the scientific methods and assumptions valid and clearly outlined? Yes.

5. Are the results sufficient to support the interpretations and conclusions? Yes.

6. Is the description of experiments and calculations sufficiently complete and precise to allow their reproduction by fellow scientists (traceability of results)? Yes.

7. Do the authors give proper credit to related work and clearly indicate their own new/original contribution? Yes.

8. Does the title clearly reflect the contents of the paper? Partly.

9. Does the abstract provide a concise and complete summary? Yes.

10. Is the overall presentation well structured and clear? Yes.

11. Is the language fluent and precise? Yes.

12. Are mathematical formulae, symbols, abbreviations, and units correctly defined and used? Yes.

13. Should any parts of the paper (text, formulae, figures, tables) be clarified, reduced, combined, or eliminated? Yes. I'll explain below.

14. Are the number and quality of references appropriate? More or less, yes.

15. Is the amount and quality of supplementary material appropriate? Yes.

**Remarks and suggestions**

1. I think that the title of the paper does not reflect the whole content of the study. The word "characteristics" is very modest, not ambitious. The part of the comparison of modelling results is not indicated in the title. It will be good if the main scientific question of the study is somehow reflected in the title.

2. The structure of the article is not the traditional but acceptable. In the introduction there is a description of the studied phenomenon but the description of the state-of-art is lacking. I would like to see here an overview of the previous studies on convective snow bands and their main results, the lack of knowledge on this topic, which will be tried to cover in this study, its main research questions and hypotheses. The main objectives and research tasks could be clearly defined in the introduction. Comparison of different model sets is not mentioned as a task in this study.

3. The second chapter has nearly the same title as the whole article. It is not justified. I recommend to combine two first chapters into one introduction. There I did not find two important terms that are closely related to convective snow bands. I suggest that they occur mostly in case of cold fronts. A description of typical synoptic situations favourable for the formation of snow bands could be much more detail in the introduction. Sea ice is also a very important factor influencing on snow bands. Different extent of sea ice has a different impact. It could be described also in the introduction.

4. In some places I found repetitions concerning methodology. The same information is presented in Table 1 and on page 5 line 31 to page 6 line 9, and on page 5 lines 14-26 and in Table 2. If there are tables with the list of criteria then the criteria need not to be listed in the text.

5. The time series of 11 years is not a long time series at all (look on page 10 line 17). It does not allow to do any climatological conclusions. Temporal variability of convective snow bands as well as of winter weather conditions in general is so large that 11 years cannot describe the climatological regime of convective snow bands. It can be clearly seen on Figure 3. I am not sure at all that the Gävle and Västervik regions are the richest of convective snow bands in the Baltic Sea coastal region at all. May-be, the Finnish or Estonian coasts have them even more. What do you think about this?

6. Page 6 line 15. There is a sentence "Most days occur in the months of November and December." Looking on Fig. 3b I am not sure that most, i.e. majority of convective snow bands occur only in these two months.

7. There is confusion with wind directions on page 6 lines 16-18. Wind direction 0-65° is not northwest but northeast direction. So, which was the most common wind direction in case of convective snow bands? Why westerly wind was not related to snow bands? I suggest that westerly wind is not related to snow bands because it brings warmer air in winter and not cold outbreaks. It is not a surprise. In my mind convective snow bands in the southern coast of the Gulf of Finland are directly related to northerly and northeasterly winds. The same

question is also on page 7 line 10. Are the snow bands really related to northwesterly and westerly winds? Please, clarify this.

8. The numeration of figures is incorrect in the section 5.2. There are referred up to 10 figures, but in fact there are 11 figures in the manuscript. Please, check the figures. There are many figures with a similar pattern (Figures 4a,b, 5, 6a,b). Are they all needed?

9. Page 11 line 10. There should be "cold Finnish land" instead of "cold Finish land".

10. Page 2 line 23. There is (Mazon et al., 2014), but in the list of references the year is 2015. Please, check it.

---

## Author Comment (AC1) · 29 Dec 2016

We would like to thank Reviewer #1 for the helpful and insightful comments on the manuscript.

1) We suggest a change of the title to "Characteristics of Convective Snow Bands along the Swedish East Coast" to be more clear about the region of interest for this study. We agree that other areas affected by Convective Snow Bands in the Baltic Sea area should be mentioned in the introduction as well. However, the focus is on the Swedish East coast because the atmospheric conditions causing lake effect snow in Sweden have clearly repeating patterns, while other areas could experience snow bands under different atmospheric conditions (e.g. other wind directions due to other coastal orientations). Convective snow bands in the Gulf of Finland have been studied

widely before. Extending our method to the entire Baltic Sea would require a different approach with more lose and generous criteria. This could lead to capturing other precipitation events, which are unrelated to convective snow bands, also in different regions and would manipulate the climatological results. This will be further clarified in the introduction.

2) A regional climate model was chosen instead of a numerical weather prediction (NWP) model to evaluate the potential of applying climate model and make climatological studies (in contrast to most previous studies being higher resolution process studies). Performing simulations at a high resolution is computationally expensive and time consuming, and therefore it is not reasonable to run a NWP model at a climatological time scale. The challenge in choosing an appropriate model for climatological studies of meso-scale phenomena is to find a balance between computational expense and accuracy of the simulated physical processes. In order to understand how precisely the RCA model performs for the atmospheric conditions associated with convective snow bands, case studies have been carried out and different setups of the RCA model were evaluated specifically for those case studies. Being aware of the benefits and weaknesses of the chosen model setup helps for the interpretation of the climatological results. The new contribution, in terms of the modelling, is to show that a relatively coarse-resolution model can be used with the potential of applying it to climatological studies. The conclusions are (and here we agree with the reviewer, that the results are not very surprising) that introducing higher resolution makes a difference and that the better SST makes a difference. This has the implication that we can make a climatology based on a regional climate model, and it would also be possible to investigate the impact of climate change on the frequency of occurrence (or distribution). Additional text will be added in the manuscript to clarify this.

3) We will clarify in the paper that the snowfall occurs not only at the coast, but also over the sea.

4) The purpose is not to make a through analysis of the agreement between remote

sensing products and modelling results. We make a sensitivity analysis on the factors in the mode (set-up, resolution etc.) influencing the results. The different set-ups of the model are compared with observed precipitation to identify problems in the model in reproducing the accurate precipitation. We agree with the reviewer that it would be very interesting to evaluate the precipitation rates based on radar products. We, however, consider this to be outside the scope of the paper. The satellite image is included merely as an illustration of the analysed situation, rather than to be used as a measure on the accuracy of the models. We would like to keep the image.

5) The aim is to define simplified conditions reflecting the local and large scale for the occurrence of precipitation related to convective snowbands. Large scale conditions are expected to be enough reflected in the wind and temperature conditions (defined over the larger area). This will be additionally discussed in the text.

6) More references will be reviewed and cited in the introduction. However, although my criteria are based on other references, they have been adjusted to the region by investigating the resulting days by hand and confirming them with satellite images. Anyhow, the criteria are logical and generally valid. Including weaker criteria in another category also gives another 'buffer zone' for a pool of days to be careful with in the dataset.

7) The data shown in the figures 4, 5 and 6 represent the accumulated hourly snowfall rates. By definition of the snowfall parameter in the model, there is no rain included. As explained in page 6, line 33 and following, the criteria in table 1 were only applied for the specific sectors (as in figure 2). The temperature differences are therefore only fulfilled for the reference sectors, not over the Gdansk region. Showing a larger area than the selected reference sector of interest helps determining other potential areas which can be affected by lake affect snow under the same conditions as the Swedish east coast experiences them. References for convective snow bands affecting the Gdansk region can be included in the revised version of this paper.
Specific comments

"The precipitation amounts connected to these snow bands are not large or exceptional, even 17 mm per 24h is too low to call extreme. (the paragraph beginning at p 6 line 22)" – Right! Probably because of the low resolution of the model. All precipitation values are underrepresented, so perhaps in reality they are extreme. However, I removed the word 'extreme' in the paper. "Why only the precipitation for the sector shown in Fig 4 is presented? Are other data missing or the days selected could not be associated with precipitation in other regions?" – We are showing a larger environment around the applied sector to see effects in other regions (such as in Gdansk). No data is missing, but we wanted to keep the focus on the Baltic Sea and not confuse the results with other precipitation areas that are not associated with convective snow bands. We could extend the area to the Gulf of Finland, that could be interesting too, but we would have to be careful with the interpretation.

---

## Author Comment (AC2) · 29 Dec 2016

We would like to thank Reviewer #2 for the helpful and insightful comments on the manuscript.

1) Following the comments by both reviewers we suggest the title to "Characteristics of Convective Snow Bands along the Swedish East Coast".

2) The second section of the paper provides a small literature review about previous studies and current knowledge about convective snow bands to the extent required to motivate the method of this paper. However, more literature will be reviewed in the introduction to give a better background on the state of art. The objective and research questions will be formulated more clearly.

[Figure]

3) The first two sections will be combined in the final paper to merge objectives and research questions better with the literature review of previous studies. The synoptic situations leading to convective snow bands can be very different. What they have in common are the strong pressure gradients over the Baltic Sea guiding cold air masses from the northeast over the warm water surface towards Sweden. Cold fronts move commonly from west to east, while snow bands only affect the Swedish east coast at strong prevailing NE winds. This unusual synoptic flow can be caused by a deep low pressure system southeast of the Baltic Sea and/or indirectly by a local high pressure developing over the cold north of Scandinavia. Since the large scale synoptic situation for convective snow bands cannot be identified by typical conditions and only the strong NE winds with small vertical shear advecting cold air matter in connection with the local conditions, I consider the present description to be sufficient. Sea ice limits the heat fluxes from the sea surface and changes the coastline. The development of snow bands requires an ice-free and open water surface. This was discussed at different parts of the paper. However, it can be clarified in the introduction once again.

4) The paragraph repeating information shown in the table will be shortened or even skipped.

5) An 11-year dataset cannot be understood as climatology and the mean results are clearly biased by single events. The phrase "climatology" was therefore not used in the paper. However, with various snow band events occurring per year, a dataset covering 11 years is able to represent qualitative distributions of affected regions. It can therefore be assumed that Gävle and Västervik experience the most intense convective snow band events in the studied area, the Swedish east coast. That also other locations at the Baltic Sea are affected under similar atmospheric conditions has been seen in the Gdansk region. The focus of this study is on the Swedish East coast because the atmospheric conditions causing lake effect snow in Sweden have clearly repeating patterns, while other areas could experience snow bands under different atmospheric conditions (e.g. other wind directions due to other coastal orientations). Convective

snow bands in the Gulf of Finland have been studied widely before. Extending our method to the entire Baltic Sea would require a different approach with more lose and generous criteria. This could lead to capturing other precipitation events, which are unrelated to convective snow bands, also in different regions and would manipulate the results.

6) These are the months with the highest frequency, this should be clarified in the text.

7) We apologize for the confusion. We mean northeast winds, which are related to cold air outbreaks from Finland or Russia. Westerly winds are usually not cold enough in winter and could either way not generate snow bands affecting the Swedish coast, because they align with the wind direction.

8) The figure numerations are corrected now.

9) The typo has been corrected now.

10) The reference has been corrected.